# Trends and Challenges in Plant Cryopreservation Research: A Meta-Analysis of Cryoprotective Agent Development and Research Focus

**DOI:** 10.3390/plants14030447

**Published:** 2025-02-03

**Authors:** Pilsung Kang, Sung Jin Kim, Ha Ju Park, Se Jong Han, Il-Chan Kim, Hyoungseok Lee, Joung Han Yim

**Affiliations:** 1Division of Life Sciences, Korea Polar Research Institute, Incheon 21990, Republic of Korea; lovesong3233@kopri.re.kr (P.K.); sungjin@kopri.re.kr (S.J.K.); hansj@kopri.re.kr (S.J.H.); ickim@kopri.re.kr (I.-C.K.); 2CRYOTECH Inc., 2F-204, 71, Mieumsandan 5-ro 41beon-gil, Gangseo-gu, Busan 46744, Republic of Korea; hajupark@cryotech.co.kr; 3Polar Science, University of Science and Technology, 26 Songdomirae-ro, Yeonsu-gu, Incheon 21990, Republic of Korea

**Keywords:** plant cryopreservation, cryoprotective agent, plant tissue culture, exopolysaccharide, polar regions, text-mining, meta-analysis

## Abstract

The stable long-term preservation of plant cells is crucial for biopharmaceuticals and food security. Therefore, the long-term cryopreservation of plant cells using a cryoprotective agent (CPA) is a crucial area of study. However, research on low-toxicity CPAs remains limited. We analyzed 1643 abstracts related to plant-cryopreservation (PCP) research published from 1967 to May 2023, spanning 56 years, from academic citation databases, with the search conducted in May 2023. Grouping these abstracts by five-year intervals revealed an increase in PCP papers until 2015, followed by a decline in the 2020s. In order to confirm the declining trend, we performed text-mining analysis using the Latent Dirichlet Allocation (LDA) algorithm, which identifies underlying topics across diverse documents to aid decision-making and classified the abstracts into three distinct topics: Topic 1, “Seed bank”; Topic 2, “Physiology”; and Topic 3, “Cryopreservation protocol”. The decline, particularly in “Cryopreservation protocol” research, is an important observation in this study. At the same time, this decrease may be due to the limited scope of Topic 3. However, we expect improvements with the development of new CPAs. This expectation is based on numerous ongoing studies focused on developing new CPAs for the cryopreservation of various animal and medical cell lines, with particular attention on polysaccharides as components that could reduce the required concentrations of existing CPAs.

## 1. Introduction

Species threatened by climate change and commercialization, along with those exhibiting high sensitivity and specific nutritional breeding requirements, require various conservation methods both within and beside their local habitats to maintain biodiversity [1,2,3,4,5,6]. Additionally, plant-derived secondary metabolites, such as phenolics, alkaloids, terpenoids, and sulfur-containing compounds, are predominantly obtained from medicinal plant cultivation for pharmaceutical purposes [7,8,9]. However, these traditional cultivation methods face significant challenges as they require vast land areas, are labor-intensive, and leave the collected plants vulnerable to diseases and natural disasters [10]. Moreover, the decrease in arable land and ongoing climate change accelerate the instability of secondary metabolite content and contribute to a shortage of plant resources [11,12]. Plant tissue culture technology has emerged as an effective method for conserving plant resources and stabilizing the production of secondary metabolites to address these issues [3,13] by enabling efficient cultivation in limited spaces. Although this technology offers significant advantages in preservation and maintaining production yields, regulations for long-term preservation may vary depending on the field, and guidelines for preservation of cell lines to maintain plant diversity are developed differently [14]. The Food and Agriculture Organization (FAO) and other organizations have published guidelines for plant cryopreservation, which are currently being refined based on existing genebank standards and practical guides [15]. In contrast, medically important cell lines developed using biotechnological methods must be stored in cell banks using LN_2_ in accordance with the current Good Manufacturing Practices (cGMP) regulations for long-term preservation [16]. Ensuring the regeneration rate of the cell lines, production of secondary metabolites, recombinant protein expression systems, and genetic stability are essential factors [17] throughout this process for the successful preservation and utilization of these valuable plant resources.

When plant cells are exposed to freezing environments, such as LN_2_, the apoplast, composed of water in the xylem–lumen space, freezes at a higher temperature than the water in the cytosol and vacuole. As a result, ice nuclei initially form in the apoplast and evolve into ice crystals that gradually increase in number [18]. The osmotic concentration of apoplastic water, which is very similar to that of pure water, causes it to freeze before the water in the cytosol and vacuole. This phenomenon helps reduce the frequency and duration of freeze-thaw cycles [19]. Physical damage to the cell membrane caused by these ice crystals can lead to dehydration, protein denaturation, metabolic disorders, and accumulation of reactive oxygen species (ROS), ultimately resulting in cell death [18,20]. Colligative additives such as DMSO and glycerol can be added to inhibit the formation of ice crystals within cells and prevent cell death under freezing conditions [21]. Additionally, cell dehydration is essential to further increase survival rates. Therefore, substances that induce dehydration outside cells, such as sucrose, ethylene glycol, and polyethylene glycol, are used either individually or in specific ratios and mixed with colligative additives [22]. These combined solutions are known as CPAs [4]. As the optimal dehydration rate and CPA tolerance vary among different cell types, methods such as slow freezing, vitrification, encapsulation vitrification, droplet vitrification, encapsulation dehydration, and cryo-plating have been used to preserve plant cells [9].

However, in addition to the selection and use of cryoprotective agents, several other challenges remain in plant cryopreservation (PCP) research. One major concern is the toxicity of high CPA concentrations, which can cause osmotic stress, increased enzyme viscosity, and DNA methylation, with the accumulation of CPAs inside cells being a primary cause of these toxic effects [9,23]. Furthermore, the successful regeneration of plant cells preserved at cryogenic temperatures depends not only on the proper selection of cryoprotective agents but also on rapid thawing and washing [9,23]. This is because slow thawing can lead to ice recrystallization, which may damage the cell membrane. To mitigate the toxic effects of CPA, strategies include adding osmotic stress inhibitors, antioxidants, and plant growth regulators to the growth medium, removing components that amplify ROS, and adjusting light conditions [23]. Additionally, various studies have suggested reducing the CPA concentration and exposure time or developing new CPAs to minimize toxicity [5,23,24,25], highlighting the ongoing efforts to improve cryopreservation techniques for plant cells.

The stable long-term storage of plant cells is crucial for the biopharmaceutical market and food security, particularly when using plant tissue cultures [26,27]. Despite the need for low-toxicity CPAs, many studies argue that research in this area is insufficient [5,23,24,25]. The purpose of this study was two-fold: to analyze the reasons behind the claim of insufficient CPA research and to propose compounds that can overcome the limitations of existing CPAs for plant cells. Therefore, we conducted a meta-analysis using text-mining techniques on abstracts collected from academic citation databases. Text-mining analysis is a method used to derive relationships between various pieces of information from numerous documents published across diverse fields, such as humanities, economics, and medicine, aiding decision-making [28,29,30,31].

## 2. Results

In May 2023, we collected abstracts related to PCP from academic citation databases (Web of Science and Scopus; Appendix A) and removed rows with empty abstracts using Excel. The clean-up and filtration processes were performed using Appendix A (Figure 1), along with various functions in RStudio, including duplicated(), filter(), str_detect(), and gsub(), which resulted in the final selection of 1643 abstracts (Figure 2). These selected abstracts were grouped into five-year intervals, and an analysis of the number of published papers revealed that publications increased until 2015s but began to decline thereafter, as shown in Figure 3. To explore the causes of these trends and underlying research patterns, a meta-analysis utilizing text mining methods was conducted.

### 2.1. Term Frequency and Bi-Gram Network Analysis

Analyzing changes in the TF used in documents enables us to infer the fundamental thematic flow of the authors [33,34]. To avoid distortion from pronouns and function words that have no meaning, we combined spacy_parse() with stop_word data from the tidytext package and a list of irrelevant words to remove meaningless terms from the analysis (Figure 2). The TF of the final selected terms was analyzed. The 1643 abstracts contained 6654 nouns and proper nouns, and the 100 most frequently used terms were generally related to PCP (Figure 4). Of these, terms related to plant tissues [seed (TF: 2253; ranking: 1/6654), shoot tip (1096; 2/6654), pollen (705; 10/6654), etc.] and agriculture [plantlet (284; 44/6654), field (220; 62/6654), fruit (215; 64/6654), etc.] were mentioned far more frequently than those related to bioengineering [transformation (67; 191/6654), transgenic (19; 573/6654), engineering (19; 582/6654), etc.] (Appendix A). We used an N-gram algorithm to examine word relationships and analyze the linguistic structure of compound words with specific meanings, addressing a drawback of the DTM: it ignores the linguistic structure within documents [34]. N-gram is a method for analyzing N consecutive words, typically focusing on two-word combinations (bi-grams) [34]. Using unnest_tokens(), we analyzed bi-grams (Figure 2) and found several word pairs appearing together more than 100 times: “cryopreservation and method”, “term and storage”, “survival and rate”, “cold and acclimation”, “cryopreserve and shoot tip”, etc. Among these, the “genetic and resource” and “plant and species” pairs were primarily used in the context of cryopreservation, while other pairs mainly described cryopreservation methods (Appendix A). Among the analyzed word pairs, a bi-gram network graph was created using word pairs with a bi-gram frequency of 40 or more (Figure 5). Because the bi-gram network forms around frequently used word pairs, it enables a more detailed understanding of the context and meaning of words [34]. In this network, we observed a primary cluster centered around the “seed” node, connecting to nodes such as “vitrification”, “method”, “cryopreservation”, “pollen”, “rate”, “germination”, “storage”, “term”, “conservation” and “plant”. This network was encapsulated by the term “plant seed cryopreservation”. Additionally, several isolated networks represented various physiological types and methodologies, providing a comprehensive view of linguistic patterns in PCP research.

After reconstructing the DTM for papers classified by topic modeling per period (Figure 2 and Figure 6), we analyzed the TF used per year for each topic, and the results are presented in Appendix A. The results revealed changes in the types and rankings of the top ten terms (Table 1). The top 10 terms in Topic 1 included the following 19 terms: pollen, seed, freezing, embryo, water, storage, germination, viability, clone, culture, temperature, desiccation, conservation, germplasm, technique, axis, survival, term, and condition. In Topic 2, the top 10 terms included 31 terms: freezing, membrane, temperature, protein, concentration, seed, chloroplast, damage, protoplast, tree, gene, water, tissue, amino acid, sugar, tolerance, seedling, winter, acclimation, culture, level, shoot, activity, stress, dehydration, condition, survival, response, change, ROS, and expression. The top 10 terms in Topic 3 included the following 23 terms: freezing, culture, shoot, suspension, callus, survival, shoot tip, plantlet, meristem, bud, sucrose, dehydration, embryo, growth, procedure, vitrification, recovery, regeneration, concentration, regrowth, step, technique, and term.

Topics 1 and 3 consistently contained 47.4% (9/19) and 34.8% (8/23) of the top 10 terms over five or more periods, respectively, whereas Topic 2 showed greater variability with only 13.0% (4/31) consistency. This suggested the introduction of new research themes and changes, particularly in Topic 2. The terms in each topic were classified into categories, such as “Organisms and Cellular Structures”, “Storage and Preservation”, “Environment and Conditions”, and “Physiological Processes and Responses”. The 31 terms in Topic 2 were primarily classified as “Physiological Processes and Responses”, “Chemicals and Metabolites”, “Organisms and Cellular Structures”, and “Environment and Conditions”, while the 23 terms in Topic 3 were mainly classified as “Organisms and Cellular Structures”, “Physiological Processes and Responses”, “Biotechnology and Techniques” and “Miscellaneous”. Additionally, bi-grams were reanalyzed to analyze the linguistic structure of compound words with specific meanings for each topic (Appendix A), and a bi-gram network was constructed based on this analysis. In Topic 1, a major network formed around the “seed” node at the center, connected to nodes such as “pollen”, “germination”, “storage”, “term”, “conservation”, and “situ”. These terms were related to seed storage and survival. Topic 2 shows several isolated networks with nodes related to stress response. Topic 3 formed networks around the “cryopreservation”, “shoot”, and “medium” nodes, consisting of terms related to cryopreservation protocols (Figure 7). These analyses provide insights into the evolving focus and methodologies of PCP research across different topics and periods.

We reviewed the literature to comprehensively analyze the frequency of various compounds used as CPAs and identified 61 additional compounds, including DMSO, sucrose, glycerol, and ethylene glycol (Table 2). In Topic 1, 12 compounds were used as CPA, with sucrose, plant vitrification solution (PVS), and glycerol being the top three. Topic 2 utilized 53 compounds, including sucrose, proline, and a PVS. Topic 3 included 30 compounds, with sucrose, PVS, and glycerol ranking the highest. Sucrose, the most frequently mentioned CPA (798 mentions), was used across all topics and remained important in cryopreservation research. Although its frequency in Topic 3 increased until the 2010s, it has since declined significantly. The plant vitrification solution (587 mentions) was prominently used in Topic 3, with usage sharply increasing in the 2010s but decreasing after the 2015s. Glycerol (290 mentions) was prominently used in Topic 3, with the frequency increasing until the 2010s but declining since the 2015s. DMSO (246 mentions) was used equally for all topics. Abscisic acid (132 mentions) was primarily referenced in Topic 3 until the 2010s, after which its frequency increased in Topic 2. Proline (94 mentions) saw an increased frequency in Topic 2 starting in the 2010s. Ethylene glycol (62 mentions) increased in Topic 2 during the 1990s but decreased after the 2000s, whereas its frequency in Topic 3 increased in the 2010s but gradually declined. The frequency of glucose (58 mentions) decreased across all topics over time. Ascorbic acid was not mentioned in Topic 1 but increased in frequency in Topic 3 during the 2010s and in Topic 2 during the 2015s. Sorbitol primarily appeared in Topic 3 but showed a gradual decrease in frequency. After analyzing the results in Table 2 by categorizing each compound according to its respective topic, it was found that research in Topic 3 primarily focused on CPAs, such as sucrose, glycerol, DMSO, abscisic acid, ethylene glycol, ascorbic acid, and sorbitol, as components of PVS. In this analysis, compounds such as sucrose, glycerol, etc., which are components of various cryoprotectant solutions, were counted multiple times when mentioned separately across different solutions. This approach was employed to accurately represent the frequency of each compound, receiving that substances like sucrose are independently mentioned in diverse cryoprotectant solutions within various research contexts. For Topic 2, proline and glucose were the main subjects of this study. In contrast, Topic 1 shows lower diversity and variability of key terms compared to other topics because cold and dry storage methods are commonly used for seed preservation [35]. However, these methods alone cannot completely eliminate contamination from insects, viruses, fungi, etc. [14]. Thus, although freezing preservation methods are also employed for long-term seed storage [35], research related to Topic 1 continues to focus on traditional CPA methodologies (Appendix A). Table 2 lists the compounds that have been applied as CPAs only in specific years, including glycine, taxol, ATP, sulfate, Brij35, cyclitol, ethylene glycol monomethyl ether, pinitol, quercitol, polyvinyl alcohol hydrogel, polyhydroxylated alcohol, formamide, thiourea, urea, iodoacetic acid, p-chloromercuribenzoic acid, thiol, erythritol, trimethylnonyl polyethoxy ethanol, o-methyl-muco-inositol, onitol, quebrachitol, dulcitol, and trimethylglycine. These 26 compounds accounted for 49.1% of the compounds applied as CPAs in Topic 2. In Topic 3, four compounds were utilized as CPAs: kaempferol 7-O-glucoside, trimethylamine oxide, floroglucinol (noted as phloroglucinol in standard nomenclature but retained as floroglucinol in alignment with its usage in the analyzed studies), and maltose. Additionally, research on trehalose biosynthesis genes, which facilitate the production of trehalose as a CPA, was also included. Together, these accounted for 16.7% of the compounds and related mechanisms used in cryoprotective applications. These substances have only been utilized in one-time studies but have been applied to CPA through specific strategies. In Topic 2, compounds such as glycine, cyclitol, iodoacetic acid, p-chloromercuribenzoic acid, thiol, o-methyl-muco-inositol, ononitol, pinitol, quebrachitol, and quercitol were used to enhance the cryopreservation effects through the individual treatment of plants based on metabolic processes.

Erythritol, trimethylnonyl polyethoxy ethanol, Brij35, and ethylene glycol monomethyl ether were also used to enhance cryopreservation effects through individual plant treatments (doc_id_doc013653, doc025335, doc025335, and doc071576; Appendix A). Compounds, such as ATP and sulfate, enhance cryopreservation in plants (doc_id_doc072836, doc073121, and doc107811; Appendix A). Taxol, polyvinyl alcohol hydrogel, polyhydroxylated alcohol, and potassium nitrate improved cryopreservation effects when combined with commonly known CPAs (doc_id_doc054360, doc116322, doc019135, doc024625 and doc078297; Appendix A). Additionally, formamide, thiourea, and urea have been proposed as alternative compounds that can counteract the cytotoxic effects of dimethyl based on molecular dynamics simulations (doc_id_doc009667; Appendix A). In Topic 3, trimethylamine oxide was used to enhance the cryopreservation effects when used alone on plants (doc_id_doc005481; Appendix A). Kaempferol 7-O-glucoside, floroglucinol, and maltose improved cryopreservation effects when combined with commonly known CPAs (doc_id_doc001427, doc004615 and doc008062; Appendix A). The trehalose biosynthesis gene is utilized to increase trehalose content within cells by inserting it into shoot tips to enhance cryopreservation effects (doc_id_doc005194; Appendix A).

### 2.2. Topic Modeling Analysis

Topic modeling is a crucial aspect of our analysis, with the IDM serving as a key tool. IDM visually represents the weights of topics and the distances between them, aiding in estimating the optimal number of topics for categorizing document data and understanding the degree of relevance of each topic [36]. When we divided the 1643 abstracts into three groups (k-means = 3), the topics were separated, indicating a low degree of correlation between them and suggesting that each topic represented a relatively distinct research area. The comparable sizes of the circles in the IDM indicated that the number of documents included in each topic was comparable (Appendix A). LDA, an unsupervised machine learning algorithm, was used to classify the topics. It calculates the probabilities of term repetition, occurrence frequency, and similar word arrangements after natural language processing [37]. This method has been used to identify “topics” and analyze their temporal dynamics and sematic content and has been successfully applied in previous studies [30,31,38]. In our study, we used the LDA() function from the topic model package, set with k-means values and Gibbs sampling methods, to classify the 1643 papers into three topics (Figure 2). The classification resulted in Topic 1 having 606 abstracts, Topic 2 having 406 abstracts, and Topic 3 having 655 abstracts. Because classified documents may have a probability of belonging to various topics, the total number of classified documents may exceed the actual number used in the analysis [34]. A review of the abstracts with high gamma values for each topic revealed distinct patterns in research focus. Topic 1 revealed a significant distribution of abstracts related to preserving seeds and protoplasts of endangered species or plants used in industry. Topic 2 primarily focused on the physiology of cryopreserved plants, whereas Topic 3 contained numerous abstracts explaining freezing methods for various plant species (Appendix A). Analysis of the top 20 words with a high probability of appearing in each topic (beta value) further supported these distinctions. Topic 1 included words such as seed, storage, water, germination, and pollen; Topic 2 featured temperature, gene, level, tolerance, and stress; and Topic 3 highlighted shoot tip, survival, culture, PVS, and sucrose (Appendix A). By reviewing the abstracts categorized by each topic (Appendix A), the words that are likely to appear frequently in each topic (Appendix A), the types and ranking changes of the main terms (top 10 terms; Table 1), and the results of the bi-gram network for each topic (Figure 7), it can be concluded that the words encompassing Topics 1, 2, and 3 are “seed bank”, “physiology”, and “cryopreservation protocol”, respectively (Table 3). We classified the three topics by year to analyze the temporal changes (Figure 6). The proportion of research related to Topic 1 (seed banks) sharply increased in the 2010s, whereas fluctuations in the research proportion for other periods remained similar. Topic 2 (physiology) showed a decreasing trend from the 1990s to the 2010s, followed by an increase from 2015 onwards. Topic 3 (cryopreservation protocol) exhibited an increasing trend from the 1990s to 2005, remained steady until 2015, and began to decline in the 2020s.

## 3. Discussion

The landscape of academic publishing has undergone significant changes in recent decades, with the number of published papers increasing more than sevenfold since 1980 [39] and publication speeds improving dramatically. This exponential growth in research output has necessitated the development of more efficient methods for processing and extracting information from the ever-expanding corpus of published text data [40]. In response to this challenge, text-mining analysis has emerged as a powerful tool that enables researchers to efficiently process and extract insights from vast amounts of textual information [28,29,30,31]. Literature reviews that use text-mining techniques have proven particularly valuable in providing comprehensive background information for research and in integrating existing knowledge resources to propose novel hypotheses [41,42]. By leveraging these advanced analytical methods, researchers can more effectively navigate the complex and rapidly evolving landscape of scientific literature and uncover patterns and connections that may otherwise remain hidden in the sheer volume of published work [31,38].

An analysis of TF across 1643 papers revealed that terms related to plant tissue types and agriculture were used significantly more frequently than those related to bioengineering (Appendix A). Furthermore, bi-gram network analysis showed that various nodes were connected around the “seed” node, forming a major network encompassing “plant seed cryopreservation” (Figure 5). These analyses indicate that PCP research has garnered more attention in agriculture-related studies than in bioengineering, highlighting a critical research gap that requires further investigation within the field of bioengineering. However, PCP remains a crucial issue in the field of bioengineering. Biopharmaceutical proteins produced in plant cells face challenges such as low productivity and glycosylation issues compared to microbial and animal cells; however, research to address these problems is actively underway [43]. Plant cell culture systems can establish production pipelines suitable for current Good Manufacturing Practice (cGMP) procedures [43], and their advantages, including safety from animal-derived viruses and low cost, are expected to increase the production of high-value biopharmaceuticals [16,43,44,45,46]. A study published in 2004 [46] emphasized the need to develop and validate standard procedures for plant cell cryopreservation. Among the 1643 papers analyzed, only one study published in 2007 met this suggestion (doc_id_doc002023; Appendix A) [47], with no similar studies conducted since. The US Food and Drug Administration (FDA) approval process may take a long time, but the limited research on PCP is also believed to be a contributing factor. As a result, only two plant-made pharmaceuticals have been FDA-approved to date: a vaccine produced in the *Nicotiana tabacum* cell suspension culture expression system (vaccine against Newcastle disease virus in poultry) and recombinant human glucocerebrosidase produced in the *Daucus carota* cell suspension culture system (for Gaucher’s disease treatment) [48,49,50] as cited in Lee et al. [51]. This phenomenon underscores the critical need for more research on PCP in the bioengineering field, particularly in relation to plant cell culture, to advance the development of plant-derived pharmaceuticals.

Topic modeling identified distinct thematic clusters, encompassing seed bank, physiology, and Cryopreservation protocol, which provide critical insights for shaping future research priorities. For example, an increased focus on CPA innovation in Topic 3 may address emerging challenges in PCP. Additionally, the diverse CPA strategies outlined in Topic 2 highlight the potential for customized approaches to enhance plant stress tolerance during cryopreservation. To assess temporal shifts in research focus, topic modeling was applied to analyze trends across 1643 abstracts (Figure 6). The 2010s witnessed significant fluctuations in research proportions for Topics 1 and 2 compared to other years, likely influenced by the establishment of the Svalbard Global Seed Vault (SGSV) in 2008. This event elevated seed banking as a critical global issue, spurring increased research on seed preservation to comply with SGSV deposit regulations. In contrast, research related to plant physiology experienced a relative decline. The SGSV employs traditional seed banking methods at −20 °C, distinct from cryopreservation. As such, its establishment is unlikely to have directly affected research on plant physiology or cryopreservation. Nonetheless, the SGSV’s emphasis on global seed preservation likely influenced research in seed preservation technologies, including cryopreservation. The downward trend in the number of papers published on plant cell cryopreservation since 2015 (Figure 3) was revealed by the decreasing proportion of studies classified under Topic 3 (Figure 6), which will be explained in the following section. By analyzing the usage frequency of various compounds in each topic, we examined the trends in the application of CPAs (Table 2). The analysis revealed the variability in the research topics based on the frequency and appearance period of the major compounds used in Topics 1, 2, and 3. In Topic 1, 12 compounds were applied as CPA, with basic cryopreservation compounds, such as sucrose, PVS, glycerol, and DMSO, dominating the top 10 TF rankings, indicating a focus on traditional CPA methodologies. Topic 2 used 53 compounds as CPA, with sucrose, proline, PVS, and abscisic acid ranking high in TF, suggesting the active application of various compounds related to plant metabolic processes in cryopreservation research. Topic 3 included 30 CPAs that frequently contained sucrose, PVS, glycerol, and DMSO. The total number of CPAs applied was highest in Topic 2, followed by Topic 3, and lowest in Topic 1. Although Topic 1 has a less diverse variety and variability of major terms than other topics, its research proportion was continuously maintained owing to ongoing issues related to food security caused by climate change [52]. The proportion of compounds applied as CPAs in specific years was higher for Topic 2 (49.1%) than for Topic 3 (16.7%). This disparity, along with the less diverse variety and variability of major terms in Topic 3 compared with Topic 2 (Table 1), suggests that research in Topic 3 was conducted within a relatively limited scope, leading to a gradual decrease in the research proportion related to PCP protocols (Figure 6). In PCP studies, the emphasis appears to be on applying technologies such as CPA, PVS2, and droplet vitrification rather than developing new methodologies. Nonetheless, recent studies [5,23,25,46,47] highlight ongoing efforts to innovate and develop new CPAs. Furthermore, cryopreservation research in biomedicine within bioengineering continues to progress [53], driven by the need to cryopreserve cells created for human therapeutic purposes according to cGMP regulations [16], as ensuring cell stability is a prerequisite [17]. As previously mentioned, the advantages and disadvantages of permeating CPAs (or colligative additives) such as DMSO and glycerol are summarized in Table 4. Additionally, ongoing research aims to mitigate the potential toxicity associated with these CPAs [54]. One approach involves combining permeating and non-permeating CPAs, including trehalose, sucrose, protein, hydroxyethyl starch, dextran, pullulan, levan, and polyvinylpyrrolidone [53,55,56,57,58,59,60]. Furthermore, because CPA toxicity remains a major issue [53], research is being conducted to explore new non-permeating CPAs derived from microbial exopolysaccharides (EPS) [6,61,62,63,64]. The activity of non-permeating CPAs from EPS is attributed to the abundant hydroxyl groups that interact with ice crystals, providing high ice-recrystallization inhibition activity [65]. Moreover, EPS in such a cryoprotective mechanism can combine with permeating CPAs to reduce their toxicity [66]. Additionally, controlling cell dehydration in freezing environments significantly inhibits intracellular ice crystal formation [67], which is consistent with the results of numerous PCP-related studies [5,23,25]. Despite these advancements, our analysis revealed that the term polysaccharide was mentioned only once as a factor for measuring the cell regeneration rate in doc_id_doc005615 (Appendix A). To date, no studies have measured the activity of non-permeating CPAs using exogenous polysaccharides. While EPS with non-permeating CPA activity is produced by various thermophiles and mesophiles [68], EPS derived from psychrophiles is particularly interesting due to its ecological characteristics, including high salt content and ability to protect surrounding organisms from freezing at extremely low temperatures [69,70,71]. Consequently, polar regions, such as the Arctic and Antarctic and their surrounding oceans, are considered optimal for isolating strains that produce EPS with antifreeze activity, presenting a promising avenue for future research on PCP. Microbially derived EPS are differentiated from plant- and animal-derived polysaccharides in terms of their efficiency and stability during production. The extraction of polysaccharides from plants and animals typically involves mechanical crushing and lipid removal during sample pretreatment, followed by extraction, purification (microfiltration, ultrafiltration/diafiltration), and formulation. In contrast, microbially derived EPS, such as hyaluronic acid produced by *Streptococcus zooepi* (KCTC 0075BP), can be designed to bypass pretreatment and extraction processes, proceeding directly to purification and formulation [72]. Although plant- and animal-derived polysaccharides have the advantage of being extracted from waste and by-products, potentially reducing environmental pollution [73], the supply and content of raw materials may be unstable, posing challenges for cGMP process validation, particularly in performance qualification (PQ) [11,16]. Conversely, microbial-derived EPS offers high production efficiency and the ability to control raw material supply and content according to market demand [60,73,74]. Given these advantages, PCP research should model from animal cell cryopreservation studies, as EPS is anticipated to serve as a crucial alternative to address the limitations of existing permeating CPAs. In animal cell cryopreservation, EPS has demonstrated the potential to mitigate the toxicity issues associated with conventional permeating CPAs. Accordingly, EPS is expected to play a vital role in plant cell cryopreservation by controlling cellular dehydration and preventing the formation of intracellular ice crystals. Furthermore, EPS derived from microorganisms offers distinct advantages, including ease of production and the ability to regulate raw material supply and composition based on market demand. In contrast, plant- and animal-derived polysaccharides often face challenges such as supply instability during production, making consistent manufacturing and quality control under cGMP standards difficult. Microbially derived EPS provides a promising solution to these limitations. Consequently, plant cell cryopreservation techniques utilizing EPS present a novel approach that could either replace or complement existing CPA technologies, thereby significantly enhancing the efficiency and stability of plant genetic resource preservation. This innovative strategy holds the potential to transform long-term plant resource preservation, with future research expected to establish EPS as a cornerstone in overcoming technical challenges.

## 4. Materials and Methods

### 4.1. Abstract Collection and Clean-Up for Meta-Analyses

The dataset for this study was obtained following the guidelines of the PRISMA 2020 statement and Cochrane Training, applying a systematic approach to searching for and identifying relevant studies to minimize selection bias [32,92]. The abstracts were collected from academic citation databases (Web of Science and Scopus) in May 2023 using the search terms related to PCP (Appendix A) for our initial collection (total 234,044 abstracts). The databases were selected for their extensive coverage of peer-reviewed scientific literature across multiple disciplines, ensuring the robustness of the dataset for analysis. To minimize bias and enhance data quality, we refined the dataset by removing duplicates and abstracts without relevant content using Excel (Microsoft, Redmond, WA, USA) and the duplicated() function in the base R package [93]. From this refined set, papers containing a second set of search terms associated with plants were selected using the filter() and str_detect() functions from the dplyr and stringr packages in RStudio, respectively (Appendix A). These additional search terms were intended to focus on specific thematic keywords relevant to the study, thereby improving dataset specificity. We further refined the dataset by removing abstracts containing irrelevant words (listed in Appendix A, created during the analysis process) using a filter() function. Finally, to ensure the inclusion of only relevant studies, two independent groups of reviewers manually excluded papers unrelated to the study topic. After combining the results from both groups, abstracts unrelated to the themes of this review (cryoprotectants, cryopreservation protocols, and their effects on plant cells) were discussed and excluded. In the end, 1643 abstracts were selected, as shown in Figure 1. Given the large number of included articles and limited resources, the authors did not conduct forward or backward citation searching.

We created a “synonyms table” for this study and used the gsub() function from the base R package to standardize terms (e.g., “cpa” to “cryoprotectiveagent”, “plant vitrification solution” to “plantvitrificationsolution”, “−196 °C” to “liquidnitrogen”; see Appendix A for full details) to eliminate result distortions caused by abbreviations and compound words. This step was undertaken to standardize terminology across abstracts from diverse sources, thereby minimizing variability and ensuring consistent analysis. After these edits, we used a refined set of 1643 abstracts for our meta-analysis, ensuring a comprehensive and standardized dataset for our study.

This study does not have a preregistered protocol.

### 4.2. Term Frequency and Bi-Gram Analysis

For text-mining analysis in RStudio, we utilized a suite of packages, including dplyr, tidyr, stringr, tidyverse, tidytext, tm, spacyr, topic models, word clouds, and ggplot2. We began by parsing the final collected abstracts using spacy_parse() to convert them into tidy text data and to extract lemmas. The text was then tokenized using unnest_tokens(), and all punctuation (e.g., periods, question marks, exclamation points, commas, parentheses, and quotation marks) using filter() and !str_detect() was removed. Next, the stop word list (stop_word data from the tidytext package; e.g., articles, conjunctions, and forms of the verb “be”) and a custom list of irrelevant words (note: created during the analysis process; e.g., journal, copyright, etc.; see Appendix A) were removed using anti_join(). This step was crucial to focus on meaningful content by excluding irrelevant or commonly used terms.

Further refinement involved filtering common verbs, adjectives, prepositions, special characters, numbers, letters, and words with two or fewer characters using the filter() function. The final selected words were organized into a document-term matrix (DTM), and the term frequency (TF) of these organized words was calculated using the count(). We used the word cloud () function to visualize the TF. A bi-gram analysis was conducted to identify significant word pairs and their contextual relationships. The frequency threshold for bi-grams (40 or more occurrences) was established through exploratory analysis of the dataset to ensure a balance between comprehensiveness and clarity. The resulting bi-gram network was visualized using ggraph() and Cytoscape (version 3.10.2), providing a comprehensive view of the word relationships within our dataset (Figure 2).

### 4.3. Topic Modeling

Using the DTM, we used the LDAvisData, LDA, and LDAvis packages to create an intertopic distance map (IDM). The Gibbs sampling iterations (g = 5000) were selected to ensure convergence of the model, and the Dirichlet parameters (alpha and eta values set to 0.02) were chosen to enforce sparsity in topic and word distributions based on prior studies in similar fields. These settings allowed for better differentiation between topics while minimizing noise. We analyzed the size and similarity of each topic by varying the number of topics (k-means values) during preliminary testing.

We utilized topic models and LDAvis packages, classifying the DTM by topic using the k-mean value determined from the IDM analysis (k = 3) and the LDA() function to create the Latent Dirichlet Allocation (LDA) model for topic modeling. This parameterization provided a clear thematic structure for the dataset, balancing model interpretability and granularity. The results highlighted key thematic patterns and relationships among abstracts (Figure 2).

## 5. Limitations

Through this systematic approach, we identified key keywords and research trends, but there were several limitations in this study. First, our study showed a global trend of using only English-written papers due to language barriers. For a more comprehensive study, future research should include PCP papers written in other languages. This will help ensure a broader and more diverse representation of the available research and mitigate the potential biases caused by focusing exclusively on English-language sources. The reliance on abstracts may exclude contextual details available in full-text articles. Moreover, the declining trend in publications post-2015 necessitates further exploration, potentially examining funding patterns, shifts in research priorities, or technical challenges in PCP.

## Figures and Tables

**Figure 1 plants-14-00447-f001:**
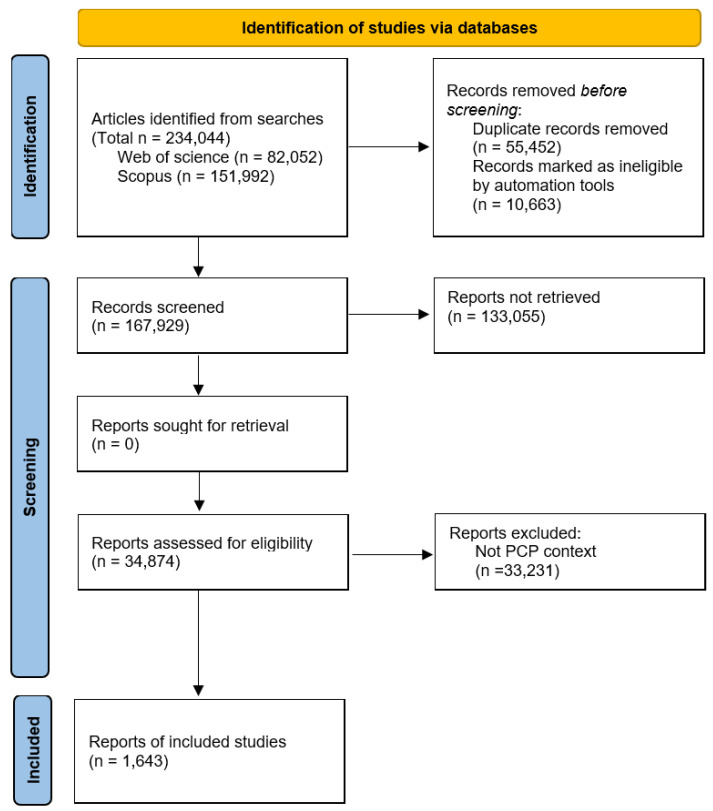
Number of the collected abstracts and cleanup process for meta-analysis. Of the 234,044 abstracts identified, 1643 met all screening criteria and were included in this study. Adapted from Page et al. [32].

**Figure 2 plants-14-00447-f002:**
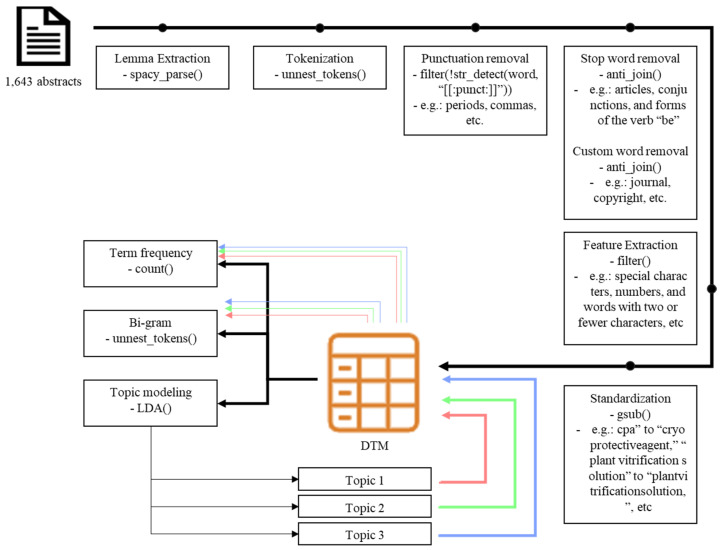
Text-mining workflow for data preprocessing and topic modeling using RStudio.

**Figure 3 plants-14-00447-f003:**
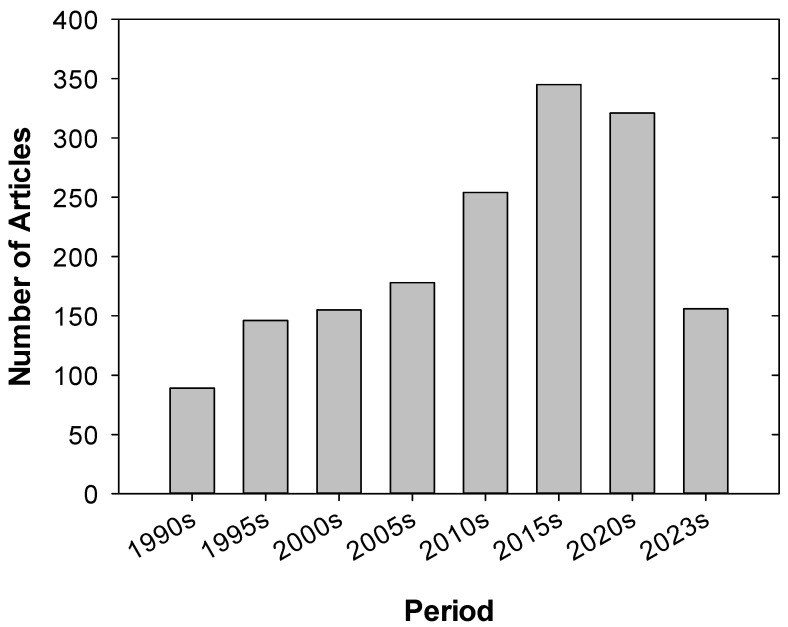
Distribution of plant cryopreservation (PCP) articles by year (N = 1663). Numbers indicate total articles published in five-year intervals (1990s: 1967–1990; 1995s: 1991–1995; 2000s: 1996–2000; 2005s: 2001–2005; 2010s: 2006–2010; 2015s: 2011–2015; 2020s: 2016–2020; 2023s: 2021–2023). The 2023s category includes articles from January 2021 to May 2023.

**Figure 4 plants-14-00447-f004:**
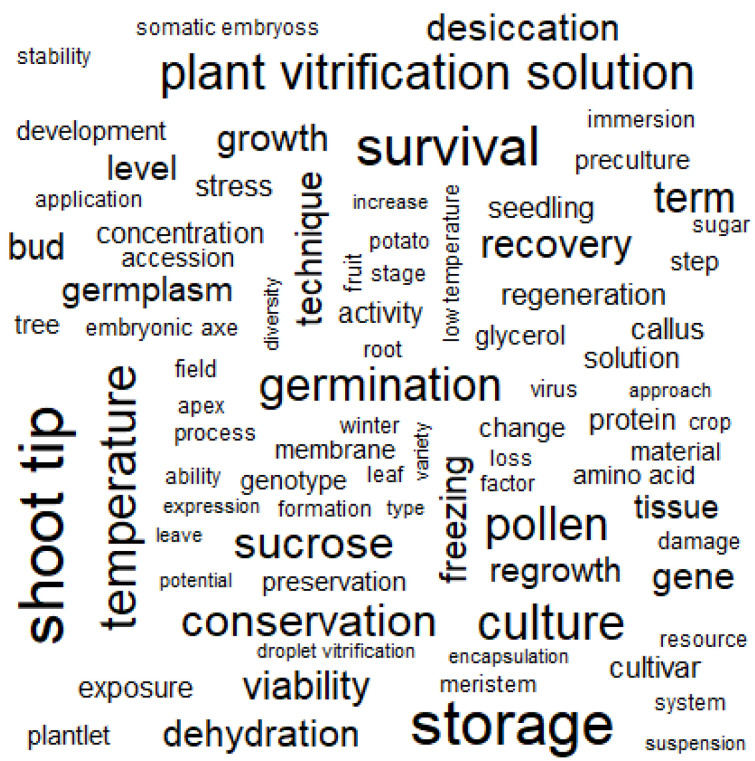
Word cloud visualization after term frequency (TF) analysis of 1643 abstracts. Word size reflects frequency, with larger words appearing more frequently in the dataset. This visualization provides an intuitive overview of key concepts and research areas within the field.

**Figure 5 plants-14-00447-f005:**
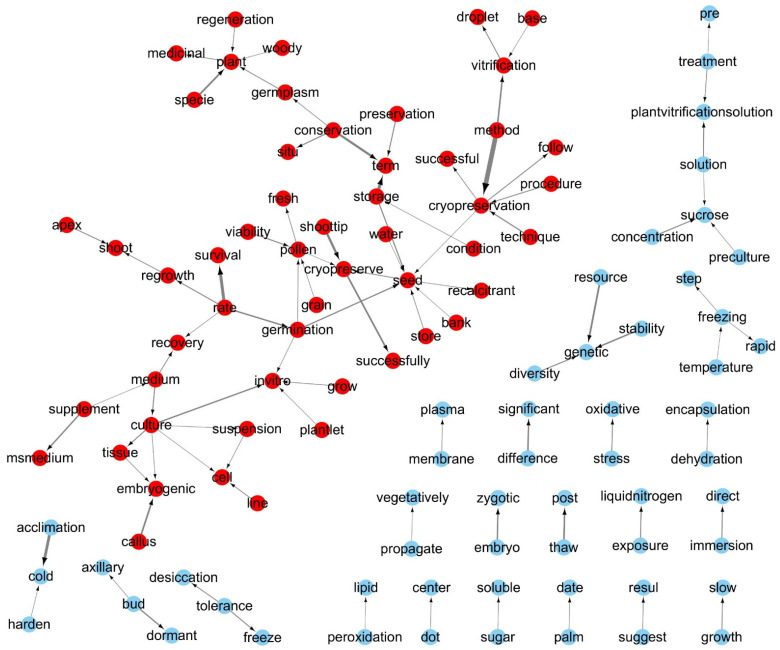
Bi-gram network of term pairs with a frequency over 40 in 1643 abstracts. Edge (arrow) widths represent bi-gram frequency, with thicker arrows indicating higher frequencies. Arrows show the directionality of bi-grams.

**Figure 6 plants-14-00447-f006:**
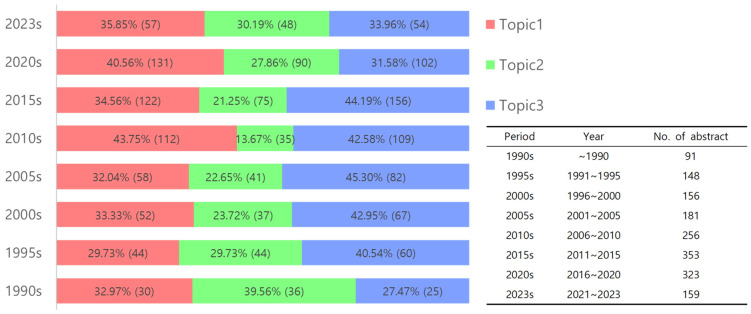
Temporal trends in topic distribution from the 1990s to 2023s. Percentages and sample sizes (in parentheses) within each segment indicate the proportion of each topic during the given period. Topic 1: Seed bank; Topic 2: Physiology; Topic 3: Cryopreservation protocol. Note: The number of categorized abstracts may exceed the number of abstracts used in the analysis, as each abstract may belong to multiple topics.

**Figure 7 plants-14-00447-f007:**
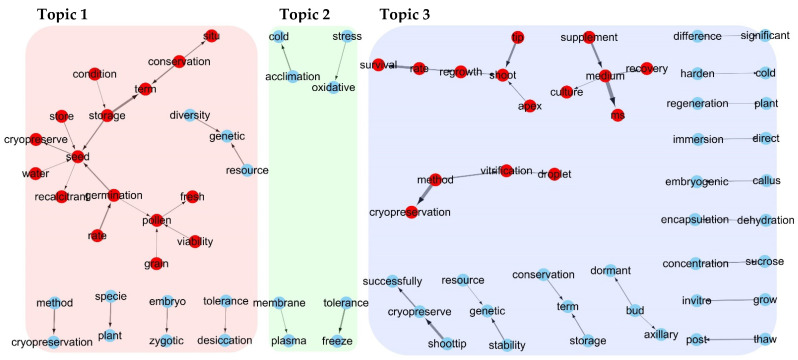
Bi-gram network of term pairs with a frequency > 40, grouped by topic. Edge (arrow) widths represent bi-gram frequency, with thicker arrows indicating higher frequencies. Arrows show the directionality of bi-grams.

**Table 1 plants-14-00447-t001:** Trend analysis of top 10 term rankings organized by topic and year.

Classification	Term	Topic 1								Topic 2								Topic 3							
		1990s	1995s	2000s	2005s	2010s	2015s	2020s	2023s	1990s	1995s	2000s	2005s	2010s	2015s	2020s	2023s	1990s	1995s	2000s	2005s	2010s	2015s	2020s	2023s
Organisms and Cellular Structures	seed	2	1	1	1	1	1	1	1	6															
	pollen	1	4	6	8	5	6	6	4																
	embryo	4	5			8	8												8						
	germplasm			8				10																	
	axis				7																				
	clone	9																							
	protoplast									9	6														
	tissue										9			10											
	tree									10															
	shoot												9					3	6	1	3	4	3	5	9
	shoot tip																	7	3	3	1	1	1	1	3
	bud																	10		5			10	9	8
	meristem																	9	10						
	callus																	5							
	plantlet																	8							
Storage and Preservation	storage	6	2	2	3	3	2	2	3																
	freezing	3	6	5	5	6		8	9	1	5	7			9			1	1	2	8	6	9		5
	conservation			7	10		3	3	2																
	viability	8	8			9	9	7	6																
	seedling											9	1												
Physiological Processes and Responses	germination	7	7	4	6	2	4	5	7																
	desiccation		10	9	4	7	7																		
	survival				9										10			6	2	4	2	7	4	3	1
	stress													5	1	1	2								
	tolerance											8	3		7	8	5								
	response															6	4								
	damage									8		6					10								
	acclimation												6												
	expression																7								
	ROS															10									
	vitrification																				5	3	6	7	10
	regrowth																					5	7	4	
	recovery																				7		5		4
	dehydration																		7	9		10		10	
	growth																		9	10					7
Environment and Conditions	water	5	3	3	2	4	5	4			8			7											
	temperature		9							3	1	1	2	1	2	5	3								
	culture	10											7					2	4	7	6	8	8	6	2
	condition								10					9											
	winter											10													
Chemicals and Metabolites	gene										3	3	4	2	5	3	1								
	protein									4	4		5	3	4	4	8								
	membrane									2	2	4			6										
	amino acid										10			8			9								
	sugar											5	10												
	chloroplast									7															
	dehydrin													6											
	sucrose																		5	6	4	2	2	2	
Biotechnology and Techniques	technique			10					8															8	
	suspension																	4							
	procedure																			8					
	regeneration																				9				
	step																					9			
Miscellaneous	term					10	10	9	5																6
	activity													4	3	2	6								
	concentration									5	7	2									10				
	level												8		8	7									
	change															9									
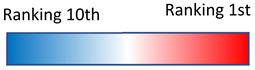																									

**Table 2 plants-14-00447-t002:** Frequency of compounds used as CPAs organized by Topic.

Compound	Classification	Topic 1								Topic 2									Topic 3							Total Frequency
		1990s	1995s	2000s	2005s	2010s	2015s	2020s	2023s	1990s	1995s	2000s	2005s	2010s	2015s	2020s	2023s	1990s	1995s	2000s	2005s	2010s	2015s	2020s	2023s	
sucrose	Sugars	6	9	2	11	22	25	33	4	11	19	14	15	7		12	14	9	51	74	97	137	187		39	798
PVS	Mixture			4	1	17	14	29	12					1	4	36	13		9	19	48	139	176		65	587
glycerol	Polyols	3	2	2		3	8	8	4	3	11		1	1	14	2		1	11	19	30	45	66	46	10	290
DMSO	Sulfoxides	8	1	1	1	6		8	1	8	5	11	1	3	15	3		19	33	16	26	30	17	22	11	246
abscisic acid	Hormones										5	7		16	15	11		1	7	19	13	22		8	8	132
proline	Amino acid										12			8	21	27			6	3		8	3	6		94
ethylene glycol	Glycols					2			1		8	1	1	1	4			2	4	4	3	11	10	7	3	62
glucose	Sugars	2	1			2	4			8	10			4	3			3	10	3	3	5				58
ascorbic acid	Antioxidants														12	14						7	10	13		56
sorbitol	Sugar alcohols		1			2					1		2					6	14	8	7	10			4	55
alginate bead	Polymers			2	3	2	3	6											14	15		1				46
trehalose	Sugars									4		5		1	4						1	10	14	7		46
polyethylene glycol	Glycols	4				6				7	2	7			1			3		3		1				34
phloroglucinol	Phenolics							1															16	12		29
antifreeze protein	Proteins												8			17										25
calcium chloride	Inorganic Salts										1				1			2				10	6			20
mannitol	Sugars			1		2				1	2			1	2				1	4	1	3				18
melatonin	Hormones															5								6	6	17
carbohydrate	Carbohydrates					2					4	9									1					16
glutathione	Amino acid														7						4	3				14
amino acid	Amino acid									4	4				5											13
glycinebetaine	Amino acid										5				4							3		1		13
glycine	Amino acid															9										9
tocopherol	Antioxidants																			2		7				9
salicylic acid	Hormones														3		5									8
taxol	Hormones										7															7
brij35	Surfactants										3	2										2				7
polyvinylpyrrolidone	Polymers										2				1							2				5
saccharide	Sugars														4		1									5
propylene glycol	Glycols														3				1						1	5
kaempferol 7-O-glucoside	Phenolics																					5				5
ATP	Nucleotides									4																4
methanol	Alcohols										1	2			1											4
lipoic acid	Antioxidants														1							3				4
sulfate	Inorganic Salts									3																3
cyclitol	Sugar alcohols											3														3
polyol	Polyols											2					1									3
trimethylamine oxide	Organics																							3		3
ethylene glycol monomethyl ether	Ethers											2														2
pinitol	Sugar alcohols											2														2
quercitol	Sugar alcohols											2														2
polyvinyl alcohol hydrogel	Polymers														2											2
polyhydroxylated alcohol	Sugar alcohols														2											2
formamide	Amides																2									2
thiourea	Urea Derivatives																2									2
urea	Urea Derivatives																2									2
floroglucinol	Organics																								2	2
trehalose biosynthesis gene	Gene																						2			2
iodacetic acid	Reagents									1																1
p-chloromercuribenzoic acid	Reagents									1																1
thiol	Proteins									1																1
erythritol	Sugar alcohols										1															1
trimethylnonyl polyethoxy ethanol	Surfactants										1															1
O-methyl-muco-inositol	Sugar alcohols											1														1
ononitol	Sugar alcohols											1														1
quebrachitol	Sugar alcohols											1														1
potassium nitrate	Inorganic Salts														1											1
maltose	Sugars																				1					1
dulcitol	Sugars																1									1
trimethylglycine	Amino acids																1									1
diethyleneglycol	Glycols																1									1
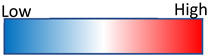																										

**Table 3 plants-14-00447-t003:** Summary table of the classification analysis.

Category	Topic 1	Topic 2	Topic 3
Number of Abstracts	606	406	655
Summary of Topic	These studies evaluate various methods for the long-term preservation of plant seeds and pollen (cryopreservation, drying, and rehydration), analyze the effectiveness of each method, preservation conditions, and changes in viability, and discuss the challenges and advantages and disadvantages of these preservation techniques in plant conservation.	These studies demonstrate that physiological responses and defense mechanisms of plants in extreme environments, particularly the accumulation of antioxidants, sugars, changes in membrane permeability, and gene expression, play a crucial role in adaptation to extreme conditions.	These studies investigate various cryopreservation techniques, including shoot tip culture, encapsulation-dehydration, and vitrification, for the long-term preservation of different plant species, such as apple root stocks, mat rush, purple-fleshed potatoes, shallots, and grapevines, highlighting the effectiveness of these methods in shoot regrowth, and genetic stability across multiple species.
Top Words (Beta Values)	Seed, storage, water, germination, pollen, etc.	Temperature, gene, level, tolerance, stress, etc.	Shoot tip, survival, culture, plant vitrification solution, sucrose, etc.
Main Term (Conclusion)	Seed bank	Physiology	Cryopreservation protocol
Supporting Evidence	gamma and Beta values (Appendix A; Appendix A); Bi-gram network (Figure 7); Top terms (Table 1)		

**Table 4 plants-14-00447-t004:** Advantages and disadvantages of various CPAs.

	Advantage	Reference	Disadvantage	Reference
Sucrose	Cryoprotection and Tolerance Enhancement	[75]	Toxicity at Higher Concentrations	[76]
	Vitrification	[76]	Non-penetration	[77]
	Reduced Toxicity		Intramolecular Hydrogen Bonding	[78]
	Basic Energy Supply			
	Dehydration			
Plant Vitrification Solution	Dehydration	[79]	Prolonged Exposure Leads to Cell Damage	[79]
	Long-term Conservation	[80]	Limited Dehydration Time	[81]
	Preservation of Cellular Structures	[82]	Unacceptably Low Regeneration	[75]
	Modifications to Reduce Toxicity	[76]		
	Vitrification			
	Penetration			
Glycerol	Osmotic Stress Neutralization	[82]	Decreased survival rate at high concentrations	[83]
	Basic Energy Supply	[76]	Limited Effectiveness	[60]
	Penetration		Lengthy Removal Process	
	Dehydration			
DMSO	Dehydration	[76]	Decreased Survival Rate at High Concentrations	[83]
	Penetration	[84]	Genetic Mutation	[76]
	Vitrification	[85]	Removal Challenges	[86]
Ethylene Glycol	Vitrification	[87]	Decreased Survival Rate at High Concentrations	[83]
	Penetration	[76]	Metabolic Concerns	[88]
	Dehydration		Species-Specific Sensitivity	[54]
Glucose	Basic Energy Supply	[76]		
	Penetration			
Sorbitol	Effective Replacement for Monosaccharides	[89]	Low Effectiveness	[76]
	Improved Post-Thaw Recovery	[90]		
	Frost Resistance	[76]		
	Dehydration			
Propylene Glycol	Vitrification	[91]	Increases the Acidity of the Organism	[88]
	Penetration	[76]		
	Dehydration			
	Penetration		Toxicity at High Concentrations	[54]
			Potential Damage to Cellular Structures	[91]
			Species-Dependent Effectiveness	

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
