# Peer review of "Trends and Challenges in Plant Cryopreservation Research: A Meta-Analysis of Cryoprotective Agent Development and Research Focus"

_plants, 2025, doi:10.3390/plants14030447_

Round 1
Reviewer 1 Report
Comments and Suggestions for Authors
See attached document

Author Response
Thank you for your feedback. We have taken your suggestions into careful consideration and made the following improvements to the manuscript.
Comments 1: Line 15. Although conservation is mentioned in the article, it should also be mentioned here as an area in which long-term preservation is important.
Response 1: Thank you for your insightful suggestion to include conservation as an area where long-term preservation is important. We have revised the manuscript accordingly to address this point. The updated text now emphasizes the importance of cryopreservation for plant cell conservation alongside its relevance to biopharmaceuticals and food security. (Page 1, line 16-18)
Comments 2: Line 20. The decline mentioned here is, in my opinion, one of the important points of the paper.
Response 2: Thank you for highlighting the importance of the decline in PCP-related papers as observed in our study. We agree with your observation and have revised the manuscript to emphasize this point. Specifically, we elaborated on the declining trend using a detailed analysis based on the Latent Dirichlet Allocation (LDA) algorithm, which classified the abstracts into three topics. The decline in “Cryopreservation protocol” research is now explicitly noted as a key finding in the study. (Page 1, lines 20-26)
Comments 3: Line 25-26. This decline being due to the “relatively limited scope of our research” didn’t seem to be emphasized in the discussion. Perhaps I missed it.
Response 3: Thank you for your insightful comment. We agree that 'relatively limited scope of our research' was incorrect, and we have revised it to clarify the point. It has been revised to “While this decrease may be due to the limited scope of Topic 3.” (Page 1, line 27)
Comments 4: Lines 35-37. This sentence doesn’t make sense. Species don’t conserve themselves.
Response 4: Thank you for your valuable feedback. We agree with your observation that the original sentence could be misinterpreted. To address this, we have revised the sentence for clarity and accuracy. (Page 1, line 36-38)
Comments 5: Line 37-39. The end of this sentence seems redundant to the beginning.
Response 5: Thank you for your insightful comment. We did eliminate redundancy and improves clarity in sentence. (Page 1, lines 39-41)
Comments 6: Line 48 and following. The discussion of regulations. The FAO and others have guidelines for plant cryo.
Response 6: Thank you for your helpful comment. We appreciate your point regarding the existence of guidelines for plant cryopreservation, such as those from the FAO. In response, we have revised the manuscript to include a more accurate reference to the regulations and guidelines governing plant cryopreservation. (Page 2, line 48-53)
Comment 7: Line 58-60. Should mention why the apoplast would freeze at a higher temp.
Response 7: Based on the reviewer's feedback, we have provided an explanation of why the apoplast freezes at higher temperatures compared to other cellular compartments, along with additional relevant references. (Page 2, line 63-66)
Comments 8: Line 77. CPA accumulation inside cells is a cause of toxicity, not a result of it.
Response 8: Thank you for bringing this to our attention. We have revised the sentence for clarified this mis-point in the manuscript. The accumulation of CPAs inside cells is indeed a primary cause of toxicity, not a result. (Page 2, lines 79-82)
Comments 9: Line 155. Is TF a part of Figure 6? This is unclear. Or is it just a part of the supplemental Tables?
Response 9: Thank you for your comment. To clarify, we added a sentence to clarify and remove any ambiguity that TF might appear as part of Figure 6. (Page 6, lines 162-163)
Comments 10: Figure 5. Very difficult to read because of the small type, especially the rectangular box/legend in the lower right-hand corner.
Response 10: Thank you for your comment. We revised Figure 5 to ensure greater accuracy by removing the small rectangular box/legend (the captions) and the relevant legend sentence. Additionally, the figure and text sizes were adjusted to enhance readability as per the reviewer’s guidance. Similarly, we improved Figure 7 in the same format, as it had the same issue. (Pages 6, 12)
Comments 11: Table 1. Very difficult to read unless increase the magnification online.
Response 11: Thank you for your valuable suggestion. We have taken your feedback into consideration, we have revised the table for better readability. (Page 8)
Comments 12: Line 211. Here “CPAs” plural is used when it should be singular “CPA”. This happens throughout the manuscript, with CPAs being used regardless of whether it is singular or plural.
Response 12: Thank you for your insightful observation. We appreciate your recommendation, and in line with your suggestion, we corrected it to “CPA” only when it was singular, based on the context.
Comments 13: Line 212 and following. The discussion on frequency of cryoprotectants. When mention that a compound is declining, is that correlated with a decline in the literature, or is it a decline in the proportion of references to the compound? Also, when a compound like sucrose is a part of other cryoprotectant solutions, is it counted twice?
Response 13: Thank you for your thoughtful question. When we mention that the frequency of a compound is declining, it refers to a decrease in its proportion of mentions within the respective topic, not necessarily a decline in the overall literature. Additionally, compounds like sucrose, glycerol, etc., which are part of different cryoprotectant solutions, are counted separately when mentioned for different solutions. This approach was adopted to more accurately represent the frequency of each compound, as they may appear in multiple cryoprotectant contexts across different studies. To address this potential confusion, we have added a clarifying sentence to the manuscript. (Page 11, lines 237-242)
Comments 14: Lines 227-228. This is confusing. “plant vitrification solution” is mentioned as being equivalent as a CPA to sucrose, glycerol, etc. Those latter compounds are components of PVS. They shouldn’t be treated as the same. If the analysis does that, it should be flagged in the discussion.
Response 14: Thank you for bringing this to our attention. We have revised the sentence for clarified this confusing expression in the manuscript. (Page 11, lines 236-237)
Comments 15: Line 231. The fact that CPAs were not as prevalent in topic 1 is no surprise, since CPAs are not generally used for seed cryopreservation.
Response 15: We agree with some aspects of the reviewer’s point. However, Topic 1 demonstrates lower diversity and variability of key terms compared to other topics, primarily because cold and dry storage methods are commonly used for seed preservation (reference [35]). Nevertheless, these methods alone cannot completely eliminate contamination from insects, viruses, fungi, etc(reference [14]). Consequently, freezing preservation methods are also employed in some cases for long-term seed storage (reference [35]). We have inserted sentences to clarify causality in consideration of the reviewer's concerns. (Page 11, lines 243-248)
Comments 16: Line 238 and following. “five compounds were applied” and the fifth is listed as “trehalose biosynthesis genes.” This is not a compound.
Response 16: Thank you for bringing this to our attention. We have revised the sentence for clarified this confusing expression in the manuscript. (Page 11, lines 255-260)
Comments 17: Paragraph beginning in Line 246. These statements seem to need references. They seem very specific.
Response 17: We greatly appreciate the reviewer's insightful query regarding our lack of a detailed explanation and we agree with the need for references. According to the reviewer's opinion, related papers have been attached and cited. (Page 12, lines 266-282)
Comments 18. Line 255. Floroglucinal should be phloroglucinol.
Response 18: Thank you for your valuable feedback regarding the nomenclature of "floroglucinol." We have revised the manuscript to clarify this point. Specifically, we have noted that "floroglucinol" is referred to as "phloroglucinol" in standard nomenclature but have retained the term "floroglucinol" to align with its usage in the analyzed studies. This clarification has been included in the revised manuscript. (Page 11, line 256-257)
Comments 19: Table 2, also very difficult to read.
Response 19: Thank you for your valuable suggestion. We have taken your feedback into consideration, we have revised the table for better readability. (Page 13)
Comments 20: Line 288. Why are preserving seeds and protoplasts grouped together? These require very different methodologies and protoplasts have nothing to do with seeds directly. They can (and usually are) derived from non-seed tissues.
Response 20: Thank you for your feedback. While we acknowledge that seed and protoplast preservation require different methodologies and are derived from different tissues, the grouping of “seed preservation” and “protoplast preservation” into the same topic in the analysis is a result of the LDA (Latent Dirichlet Allocation) topic modeling approach. LDA identifies abstract patterns by considering the relationships between topics in the text. In the context of cell preservation of endangered or industrially important plants, the research flow reflects how these two areas could be addressed together, which is why they were grouped into the same topic.
Comments 21: Line 328 and following. “These analyses suggested that PCP research has drawn more interest from agriculture-related studies than from bioengineering.” Then there is a discussion of work in bioengineering, which is fine. It might be mentioned that this is a critical gap in research that needs to be filled.
Response 21: We appreciate the reviewer’s insightful comments and have revised the manuscript to emphasize the important research gap that needs to be addressed in biotechnology and agriculture, as suggested. (Page 17, lines 354-356)
Comments 22: Line 341. It’s not clear what “it” refers to. Is it the lengthy approval process or something mentioned before that?
Response 22: Thank you for your comment. We clarified the sentence to make it clear that both the lengthy FDA approval process and the limited research on plant cryopreservation contribute to the small number of FDA-approved plant-based pharmaceuticals. (Page 18, lines 367-372)
Comments 23: Line 357. Since the Svalbard seed bank stores seeds in conventional seed banking (-20oC) rather than in cryopreservation, would this have affected cryopreservation research at the time?
Response 23: Thank you for raising an important point regarding the establishment of the Svalbard Global Seed Vault (SGSV) and its relationship with the cryopreservation study. To provide a clear and reasonable clarification, we have added new sentences and revised the relevant sections. (Page 18, lines 380-391)
Comments 24: Line 362. Should this be “was revealed by” rather than “was due to”?
Response 24: Thank you for your suggestion. By following the reviewer's advice and changing “was due to” to “was revealed by,” the meaning that the trend was uncovered through data analysis has been emphasized. (Page 18, line 392)
Comments 25: Line 367 and following. It’s not surprising that fewer CPAs are mentioned in topic 1 since CPAs are not generally needed for seed cryo.
Response 25: We agree with the reviewer's point. However, Comments 15, there is already an expression that aligns with the reviewer's advice: “Topic 1 shows lower diversity and variability of key terms compared to other topics because cold and dry storage methods are commonly used for seed preservation [35]. However, these methods alone cannot completely eliminate contamination from insects, viruses, and fungus, etc [14]. Thus, although freezing preservation methods are also employed for long-term seed storage [35], research related to Topic 1 continues to focus on traditional CPA methodologies (Figure S1).” Please refer to it. (Page 11, lines 244-249)
Comments 26: Line 374. State that Topic 1 is lowest and Topic 2 the highest.
Response 26: Thank you for your comment. We have modified the expression in the sentence to align with the reviewer's suggestion. (Page 18, lines 403-404)
Comments 27: Line 381 and following. The “gradual decline in research” might be due to the fact that protocols were found that are widely applicable and so much of the literature is geared toward the application of those techniques (the CPA PVS2 and droplet vitrification) than in developing new methods.
Response 27: Thank you for the insightful advice that our reviewer pointed out. Therefore, we have added sentences and relevant references that correspond to additional explanations in Page 18, lines 412-415 following.
Comments 28: Line 392. Exopolysaccharides should have the abbreviation in parentheses after the first mention.
Response 28: Thank you for your suggestion. We have added the abbreviation (EPS) in parentheses after the first mention of exopolysaccharides as per your advice. (Page 19, line 425)
Comments 29: Line 409 and following. This is an interesting point, but does it warrant a whole paragraph, when it’s not directly related to the results of the study? If it’s filling a gap, that should be mentioned. Also, it is the last paragraph of the discussion and there is no summary of the overall importance of the results of this study, which would be helpful.
Response 29: This paragraph discusses the advantages of EPS, which are not directly related to the results of this study. Therefore, based on the reviewer's advice, we have merged this paragraph with the previous one to enhance the relevance and clarity of the research findings and included a section summarizing the overall significance of the study. (Page 19, lines 454-470)
Reviewer 2 Report
Comments and Suggestions for Authors
The author did a huge literatures search and analysis on plant cryopreservation research by bioinformatic method from large time span (56). It is a useful work. There are several suggestions can be considered:
please revised the abstract and add some information about not only reference catalogue but also references content summary. That is, in the result section, the author should summary some important references content and give a table or description about the advantage and disadvantage of main cryoprotective agent or tell the reader what kind of cryoprotective adapt to what kind of material or tissue. and tell reader clearly what is the challenges in plant cryopreservation research? beside cryoprotective agent, other factors?
Author Response
Thank you for your feedback. We have taken your suggestions into careful consideration and made the following improvements to the manuscript.
Comments 1: please revised the abstract
Response 1: Thank you for your comment. We have revised the abstract as per your suggestion to improve clarity and coherence. (Page 1, lines 15-31)
Comment 2: add some information about not only reference catalogue but also references content summary.
Response 2: Thank you for your suggestion. In response, we have replaced the content under the 'research focus' row in Table 3 with a 'summary of topics' to include the reference content summary as requested. (Page 16)
Comments 3: That is, in the result section, the author should summary some important references content and give a table or description about the advantage and disadvantage of main cryoprotective agent or tell the reader what kind of cryoprotective adapt to what kind of material or tissue.
Response 3: Thank you for your valuable suggestion. In response to your comment, we have revised the manuscript to include a summary of the advantages and disadvantages of permeating CPAs, also known as colligative additives, such as DMSO and glycerol. These revisions are presented in Table 4 for clarity and ease of reference. (Page 18, lines 418-421 and Page 20)
Comments 4: and tell reader clearly what is the challenges in plant cryopreservation research? beside cryoprotective agent, other factors?
Thank you for your insightful comment. In response, we have clarified the challenges in plant cryopreservation research by highlighting not only the selection and use of cryoprotective agents but also other factors, such as toxicity due to high CPA concentrations, the need for rapid thawing and washing, and strategies to mitigate these challenges. These revisions are reflected in the updated manuscript. (Page 2, lines 78-88)
Reviewer 3 Report
Comments and Suggestions for Authors This study investigates the importance of the long-term stable preservation of plant cells in biopharmaceuticals and food security, as well as the current state of research on low-toxicity cryoprotective agents (CPAs). By analyzing 1,643 abstracts related to plant cryopreservation (PCP) published between 1967 and May 2023, the study explores trends and thematic changes in PCP research. It highlights that the decline in research on cryopreservation protocols (Topic 3) may be attributed to the limited scope of the study, while also envisioning the development potential of novel CPAs, particularly those incorporating polysaccharides, to address gaps in the field of low-toxicity CPAs. The overall presentation, including clear and aesthetically pleasing figures, effectively conveys information with comprehensive and rigorous content. However, the authors should further consider the following suggestions.1.Keywords The term Polar regions is mentioned infrequently in the text, appearing only once at line 405. It is therefore not suitable to include it as a keyword.
2.Introduction Lines 58–60: Which reference supports this conclusion? Please provide the specific citation. Lines 90–93: As the final paragraph of the introduction, it should focus on introducing your research objectives and goals, rather than explaining the methods used.
3.Results In Figure 1, why is before screening italicized? In Figure 4, what is the difference between the horizontally and vertically written words? Why are some words written vertically? Lines 129–132 and 140–142: The results section should not explain why certain decisions were made; this is more appropriate for the discussion section. Also, avoid excessive citations in the results section. The same issue appears in lines 275–280. Line 156: The mention of "Table 1" appears here. Why not place the table on page 7 for better readability? Lines 246–257: Which figure is being discussed here? Please clearly label it.
4.Discussion Lines 310–323: Please revise this section. The discussion is not a review; it should focus on your experimental results and compare them with those of others. Lines 351–375: This section includes excessive descriptions of your own results, making the information overly dense and lacking sufficient comparisons with the findings of others.
5.Materials and Methods Lines 433–434: The methods section does not need to explain the advantages of selecting these databases. Similar issues appear in lines 454–455.
6.Limitations Line 498: Please ensure proper indentation at the beginning of this line.
7.References Pay attention to the format of references, particularly for items 7 (Volume), 32 (pp.), and 71. Ensure uniform capitalization in the titles of references. For example, in references 4 and 5, some titles have only the first word capitalized, while others do not. Maintain the italicization of scientific names, as in references 17 and 59.
8.Conclusion It is recommended to include a conclusion that summarizes and synthesizes the key findings of the study.
Author Response
Thank you for your feedback. We have taken your suggestions into careful consideration and made the following improvements to the manuscript.
Comments 1: In lines 90-93, at the end of the abstract, there should be a focus on introducing your research work and objectives.
Response 1: Thank you for your helpful suggestion. In response, we have revised the manuscript to include the requested explanation of text-mining analysis, stating that it is a method used to derive relationships between various pieces of information from numerous documents across diverse fields, aiding decision-making. This addition is reflected in the updated version of the manuscript. (Page 1, lines 22-24)
Comments 2: In line 156, there is a reference to "Table 1"; why not place the table on page 7?
Response 2: Thank you for your suggestion. The figures and tables were inserted in the order they are mentioned in the manuscript, so 'Table 1' was moved to the back.
Round 2
Reviewer 2 Report
Comments and Suggestions for Authors
The revised version improved greatly. The author revised and made implement carefully, So I have no addition suggestion but please to check carefully the format of references and unify them the whole text.
Author Response
Comment: The revised version improved greatly. The author revised and made implement carefully, So I have no addition suggestion but please to check carefully the format of references and unify them the whole text.
Response: Thank you very much for your positive feedback and kind comments. We greatly appreciate your recognition of the improvements made to the revised version. In response to your suggestion, we carefully reviewed the reference list and ensured that all entries, including books, reports, and scientific names, adhere strictly to the MDPI reference formatting guidelines. We have unified the format throughout the manuscript to maintain consistency and accuracy. Thank you once again for your valuable feedback, which has significantly enhanced the quality of the manuscript.